# Arsenic(III)-induced oxidative defense and speciation changes in a wild *Trametes versicolor* strain

Yuhui Zhang[1], Xiaohong Chen[1,2,3], Chenyu Wang[1], Zhilan Xia[1,2,3], Ke Xiao[4]*, Ling Xie[1,2,3]*

**1** Horticulture College, Hunan Agricultural University, Changsha, Hunan, P. R. China, **2** Key Laboratory for Vegetable Biology of Hunan Province, Changsha, Hunan, P. R. China, **3** Engineering Research Center for Horticultural Crop Germplasm Creation and New Variety Breeding, Ministry of Education, Changsha, Hunan, P. R. China, **4** Institute of Agricultural Product Processing, Hunan Academy of Agricultural Science, Changsha, Hunan, P. R. China

☯ These authors contributed equally to this work.
* shirring2003@163.com (LX); 908119362@qq.com (KX)

**Data Availability Statement:** All relevant data are within the paper and its Supporting information files.

**Funding:** This work was supported by the Natural Science Foundation of Hunan Province

## Abstract

Oxidative defense or arsenic(As) changes exhibited by *Trametes versicolor* in response to toxicity under As stress remain unclear. In this study, after internal transcribed spacer identification, a wild *T. versicolor* HN01 strain was cultivated under 40 and 80 mg/L of As$^{III}$ stress. The antioxidant contents by multifunctional microplate reader and the speciations of As by high performance liquid chromatography in conjunction with inductively coupled plasma mass spectrometry were examined to explore the detoxification mechanisms. The results demonstrated this strain could tolerate As concentration of 80 mg/L with a bio-enrichment coefficients of 11.25. Among the four antioxidants, the activities of catalase, superoxide dismutase, and glutathione in the As-stress group at 80 mg/L improved by 1.10, 1.09, and 20.47 times that of non-stress group, respectively. The speciation results indicated that As$^V$ was the dominant species in the hyphae of *T. versicolor* regardless of no-stress or As-stress. The detoxification mechanisms of this strain were involved alleviating the toxicity by increasing the activities of antioxidants, especially glutathione, as well as by converting As$^{III}$ into As$^V$ and other less toxic As species. *T. versicolor* could be used as a bio-accumulator to deal with As exposure in contaminated environments based on its extraordinary As tolerance and accumulation capacities.

## Introduction

*Trametes versicolor* (L.) Lloyd, a popular medicinal fungus called "yunzhi" in China and "turkey tail" in the western world, has a long history of use as a complementary medicine or adjuvant in some Asian countries. Its health-promoting medicinal value as traditional Chinese medicine dates back at least 2000 years [1]. Polysaccharide K from *T. versicolor* is used in conventional chemo- or radiation cancer therapies in Japan [2–4]. In addition, its capacities to

[2020JJ4366 to L. X.]. The funder had no role in study design, data collection and analysis, decision to publish, or preparation of the manuscript.

**Competing interests:** The authors have declared that no competing interests exist.

degrade lignin, remove dyes, and even generate oil from waste have garnered much attention [5, 6]. Some scientists have found that the *T. versicolor* strain has the potential capacity to enrich zirconium and selenium [7, 8]. Nevertheless, whether *T. versicolor* can show accumulation and toleration capacities towards toxic metals or metalloids and the extent of these capacities have been not reported in the literature.

As a metalloid, arsenic (As) is a major cause of life-threatening degenerative diseases affecting humans such as cancer, liver injury and inflammatory diseases [9–11]. It also enters the food chain and becomes a major threat to food safety [12]. As not only poses a huge threat to human health and ecological safety but also affects plants and fungal growth due to organism uptake and transport into water [13]. For plants and fungal growth, As is non-essential, interferes with various metabolic processes, and causes physiological and morphological disorders leading to reduced growth or death. At a low concentration, As primarily inhibits biomass accumulation [14, 15]. When As accumulates to 10–20 mg/kg in plants or fungi, it hampers critical metabolic processes, and this can lead to organism death [16]. However, a minority of fungi and aquatic plants display a contradictory phenomenon. When the As accumulation concentration in these organisms reaches hundreds of mg/kg or higher, their growth is not significantly inhibited and they survive. These organisms include arbuscular mycorrhizal fungi [16], *Elaphomyces sp*. [17], and *Sargassum piluliferum* [18]. The extraordinary traits of these organisms have sparked a flurry of research regarding As bioremediation (phytoremediation and mycoremediation). Compared with phytoremediation, a plant-based technology for the removal of toxic heavy metals, mycoremediation (bioremediation using microorganisms) is a more promising method because of its low cost, short remediation period, and high efficiency [19]. Scientists have studied mycoremediation for a few decades and have reached a common consensus that some limitations, such as the selectivity of fungi and the tolerance to heavy metals need to be solved before experimental mycoremediation methods can be turned into realities [20]. There is little possibility of these microorganisms being applied to As removal until researchers find a strain that simultaneously has high accumulation and tolerance capacities for As. Accordingly, although some species with As high- or hyper-accumulation have been reported, only *Trichoderma sp.*, as a heavy metal tolerant organism, has been extensively exploited in agriculture for environmental mycoremediation [21]. Hence, a crucial key for application is to find an extraordinary species or strain with high As accumulation and tolerance capacities.

The following question is raised in light of the above: what strategies did organisms evolve to protect against the toxicity of metals or metalloids? It should be noted that the toxicity caused by the effects of oxidative stress is ubiquitous and agreed upon, but the exact molecular mechanisms remain unclear. Under heavy metal stress, to maintain normal life activities, the antioxidant defense system (antioxidants) in organisms plays an important role that primarily includes enzymatic and non-enzymatic systems [22]. The enzyme systems include a variety of antioxidant enzymes such as superoxide dismutase (SOD) and catalase (CAT). The non-enzymatic systems include reduced glutathione (GSH), malondialdehyde (MDA), and ascorbic acid [23]. An active detoxification mechanism developed by plants, algae, and fungi to avoid heavy metal poisoning that involves GSH and peptides synthesized at the expense of GSH, including the phytochelatins (PCs), has been widely reported [24]. However, questions have been raised about whether *T. versicolor* adjusts antioxidant activities to deal with As toxicity.

Currently, the As control standard primarily refers to the total As in China and other countries. However, this is not scientific enough to evaluate the risk of As exposure based only on the total As contents. There are great differences in the toxicities of different As species, that is, their toxicity depends on the chemical forms and oxidation states (speciation) [25]. Notably, As chemicals exist in the form of organic and inorganic species. Inorganic As has two primary

oxidation states, trivalent arsenite (As III) and pentavalent arsenate (As V). As III is 60 times more toxic than As V [26]. In general, organic As compounds are significantly less toxic than inorganic As compounds. Methylated species, such as monomethylarsonic acid (MMA) and dimethylarsinic acid (DMA), have been reported to be much less toxic than As V and As III [27]. Arsenobetaine (AsB) and arsenocholine (AsC) are considered to be nontoxic. Currently, approximately 25 different As species have been identified in water, with As III, As V, MMA, DMA, AsB, and AsC found in mushrooms [25, 28]. According to some studies [25, 29], the toxicities conform to the following order (highest to lowest): As III > As V > MMA = DMA > AsC > AsB. The detection of As species has been applied in toxicological risk assessments of *Cordyceps sinensi* and *Ophiocordyceps sinensis* [30, 31]. High performance liquid chromatography (HPLC) in conjunction with inductively coupled plasma mass spectrometry (ICP-MS) because of high sensitivity and a wide linear dynamic range [25] has emerged as one of the best speciation analysis techniques. HPLC ICP-MS is used in this study to examine the compositions of the As forms in *T. versicolor* and further understand the detoxification methods from the perspective of compound transformations.

Some plants or fungi with extraordinary heavy metal tolerance have received increasing attention and trigger further study about detoxification mechanism. After a wide *T. versicolor* identified by internal transcribed spacer (ITS) and incubated under different concentration of As stress, we focused on its tolerance and accumulation capacities of As, its detoxification mechanisms based on antioxidants and speciation analysis to unlock its bio-accumulator and mycoremediation potentials.

## Materials and methods

### Strain, chemicals and medium

The *T. versicolor* HN01 strain (CCTCC M 2021010)obtained by tissue isolation of the sporocarp of a wild *Trametes species* from decaying wood of a willow tree in Ju zi zhou tou (Changsha, Hunan Province, China), was used in this study. This strain was stored in the China Center for Type Culture Collection, Wuhan University, Wuhan, China. The isolated tissue was incubated in the dark at 24°C for 10 days with potato dextrose agar (PDA) as medium.

After NaAsO$_2$ (CAS7784465, Sigma-Aldrich, USA) been pre-dried to achieve constant weight, 8.6699 g sample was dissolved in 100 mL of deionized water, 50 g/L of As III solution was prepared and stored at 4 °C.

PDA containing 12 g/L potato extract, 20 g/L glucose, and 20 g/L agar (Solarbio, China) was utilized as incubation medium. Deionized water was used during all experiments. The As concentration was controlled in PDA and water(As $\leq$ 1 mg/kg, dw).

### ITS identification and the phylogenetic tree

Freshly collected sporocarp of the HN01 samples was surface sterilized using 70% alcohol, cut with a sterilized scalpel longitudinally, and a small piece of tissue was collected from the inner core. The isolated tissue fragment was placed on the PDA with cellophane in a petri plate and incubated in the dark at 24°C for 15 days. The mature hyphae were transferred and cultured repeatedly for another 15 days to obtain pure hyphae.

The mature hyphae on the cellophane were scraped using a blade, placed into a mortar, and ground into a powder with added liquid nitrogen. A total of 50 mg hyphae was weighed and placed into a centrifuge tube for DNA extraction (fungal gDNA isolation kit, BIOMIGA Inc., USA). ITS1 (5 '–TCCGTAGGTGaACCTGCGG–3') and ITS4 (5 '–TCCTCCGCTTATTGA TATG–3') were used as the two primers for the PCR amplification. The amplification system included 25 μL PCR Mix (Vazyme Biotech Co., Ltd. China), 2 μL ITS1 (10 μmol/L), 2 μL ITS4

(10 μmol/L), 1 μL template, and 20 μL dd $H_2O$. The PCR (ABI VeritiPro PCR, Thermo Fisher Scientific, USA) reaction process consisted of the initial denaturation at 98˚C for 2 min, 35 cycles (including denaturation at 98˚C for 10 s, annealing at 54˚C for 10 s, and extension at 72˚C for 10 s), and extension at 72˚C for 5 min. The PCR products were sent to the Hunan Qingke Biological Company for DNA sequencing. The ITS sequences were searched by Blast and compared with the reference sequences at the GenBank of the National Centre of Biotechnology Information (NCBI) to facilitate their identification. The ITS data matrices generated were then subjected to a phylogenetic analysis using MEGA X.

## As stress and incubation conditions

The hyphae of *T. versicolor* HN01 were monitored on a PDA medium supplemented with three As $^{III}$ concentrations (0, 40, and 80 mg/L). Each treatment was conducted 20 times. The transferred hyphae block (6 mm) was then inoculated on the new PDA with cellophane for 10 d at 24˚C. After 10 d, the hyphae on the cellophane were collected and weighed for calculating the hyphae inhibition rate (HIR).

HIR (%) = $(W_c − W_a)$ / $W_c$× 100%, where $W_c$ (g) represents the weight of the fresh hyphae in the control (CK) group; and $W_a$ (g) represents the weight of the fresh hyphae cultivated under As stress. A high HIR value showed that the hyphae growth was strongly inhibited.

## Antioxidant activity determinations

Fresh hyphae were treated referring to Dilna Damodaran's method prior to observation using a scanning electron microscope (SEM, JSM-6380LV, Japan) for morphological analysis [20].

The activities or content of antioxidants in the fresh hyphae from the blank and arsenic stress groups were determined using a multifunctional microplate reader (SuPerMax 3000AL, Flash, China). The experiments were conducted according to the operation steps in the instructions of the reduced GSH Detection Kit (ZC-S0329), the MDA Assay Kit (ZC-S0343), the SOD Detection Kit (ZC-S0350), and the CAT Detection Kit (ZC-S0351) from the Shanghai Zcibio Tech. Co. Ltd. The definition of an enzyme activity unit is described as follows: (1) SOD (U/g, fw): SOD activity when the inhibition percentage was 50% in the xanthine oxidase coupled reaction system; (2) CAT (U/g, fw): CAT activity from 1 g hyphae to degrade 1 nmol $H_2O_2$ in a minute; (3) GSH (μg/g, fw): GSH content in 1 g hyphae; (4) MDA(nmol/g, fw): MDA content in 1 g hyphae. Each treatment was repeated three times.

## ICP MS determination

Fresh hyphae from three replicates of each treatment were harvested, dried at 60˚C for 24 h, and ground. Approximately 0.2 g of a dry sample was weighed and then digested with 7 mL $HNO_3$ and 1.5 mL $H_2O_2$ via a microwave-assisted digestion system (MARS6 microwave digestion instrument, CEM, USA). The contents of As were then determined using the ICP-MS technique (ICP MS spectrometer, Agilent 7700x, Japan) [32]. The primary working conditions of the ICP-MS in this study referred to Chen's report [33] with some parameters improved as follows: carrier gas flow: 0.8 L/min; makeup gas flow: 0.3 L/min; He gas flow: 4.3 L/min; monitored ion: $^{75}$As and $^{35}$Cl; integration times: 1.0 s and 0.1 s; replicates: three. The standard curves were separately constructed using the absorbance values from standards of known element concentrations.

The linear equation for the total As was obtained using Y = 0.0027*X + 2.09 (calibration range 0.1–100 μg/L, $R^2$ = 0.9995).

Bio-enrichment coefficients (BCF) to As were further calculated based on ICP-MS results. (BCF) to As = $C_h$ / $C_m$. $C_h$ (mg/kg, dw) represented As concentration in the hyphae of *T*.

*versicolor* and $C_m$ (mg/kg, dw) represented As concentration in the PDA medium after the hyphae harvested. BCF to As > 1 demonstrated this fungus had a relatively strong ability to accumulate As. BCF $\leq$ 1 was meaningless.

## As speciation analysis using the HPLC ICP-MS

**Reagents.** All solutions were prepared with ultrapure water that was obtained from a Milli-Q water purification system (Millipore, USA). The As element standard solution (GBW08611, 1000 μg/ml), As [III] (GBW08666, 75.7±1.2 μg/g), AsC (GBW08666, 28.0±1.1 μg/g), As [V], MMA, DMA, AsB were utilized as the mixed solution standard substance (GBW082204, 1.01±5 μg/g). [7]Li, [89]Y, [59]Co, [140]Ce, [205]Ti were utilized as the tuning solution (Agilent, Part # 5185–5959). [6]Li, [45]Sc, [72]Ge, [115]In, [209]Bi, [175]Lu, [103]Rh were utilized as the internal standard solution (Agilent, Part # 5188–6525).

**Freeze-drying for pretreatment.** Fresh hyphae from five replicates of each treatment were harvested, ground using a tissue homogenizer, freeze-dried in a vacuum freeze-drier (Thermo Savant, USA), and ground again into powders.

**Microwave-assisted digestion.** A total of 0.1–0.2 g of the sample powder was weighed precisely, placed in a microwave digestion tank with 10 mL of a 1% $HNO_3$ solution (v/v) added, and then underwent a digestion process under the following conditions: heating time of 5 min and digestion for 15 min at 90°C and 1000 W. After cooling down, the upper layer of the digestive solution was transferred to a 50 mL centrifuge tube. The solution was obtained after the residue was treated with ultrapure water several times, combined with the supernatant above, and centrifuged at 8000 r/min for 10 min. A total of 5 mL of n-hexane was added, and the solution was whirlpooled for 30 s and centrifuged at 8000 r/min for another 10 min. The supernatant was obtained, filtered through a 0.22 μm organic filter membrane, and purified using a chromatographic column prior to injection.

**As speciation analysis.** Samples were analyzed within 24 h using HPLC ICP-MS (Liquid chromatography, Agilent 1260, Germany). The chromatographic columns consisted of a PRP-X100 anion exchange column (250 mm × 4.1 mm, 10 μm, Agilent Technologies, Japan). The working conditions of the HPLC are shown as follows: 40 mmol/L $(NH_4)_2CO3$—methanol (99:1, v / v) as the mobile phase A; 0.5 mmol/L $(NH_4)_2CO3$—methanol (99:1, v / v) as the mobile phase B; pH value adjusted to 8.5 with formic acid or ammonia water; injection volume at 100 μL; flow rate at 1.0 mL/min; column temperature at 25°C; collection time of 15 min; gradient elution as the elution method. This gradient elution profile began at 0:100 (A:B), increased linearly up to 100:0 in 1 min, remained constant for 9 min, decreased linearly down to 0:100 for 2 min, and maintained constant for 3 min. Species in the samples were identified using the standards of As [V], As [III], MMA, DMA, AsC, and AsB and quantified using external calibration curves with the peak area of each standard.

## Statistical analysis

All statistical tests were conducted using with PASW 18.0 Windows and Origin 2021 software. Values are presented as mean ± SD. Data was received a homogeneity analysis by Homogeneity of Variance Test, and then statistical significance was determined by ANOVA analysis. *p* values <0.05 are considered as statistical difference, *p* values <0.01 as significant difference and *p* values <0.001 as extremely significant difference.

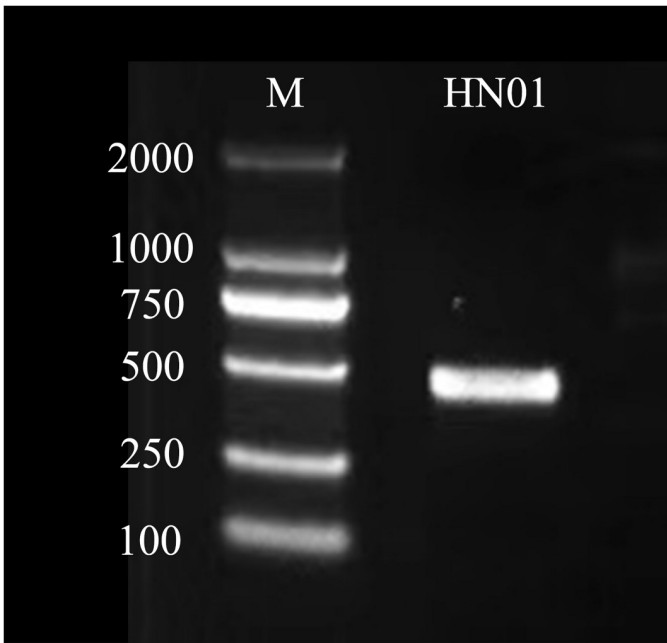

**Fig 1. Agarose gel electrophoresis of HN01 strain.**

## Results

### ITS molecular identification and phylogeny of the *Trametes* species

The ITS DNA fragment of the specimen showed a single clear band in the gel imaging system without tailing, and the length was approximately 527 bp (Fig 1 and Fig in S1 Raw images) and the value of OD260/OD280 was about 1.9. The sequencing results of the HN01 samples were compared with other *T. versicolor* databases downloaded by the NCBI to construct a phylogenetic tree, and *Ganoderma lucidum* and *Lonicera japonica* were selected as the outgroups. As can be seen from Fig 2, four strains (HN01, MW165835.1 *T. versicolor*, MK269119.1 *T. versicolor*, and AB811860.1 *T. versicolor*) formed a clade with a 100% bootstrap value. Based on phylogenetic analysis shown in Fig 2 combined with the morphological characteristics, it could be basically identified as *T. versicolor*.

### Hyphae growth of *T. versicolor* and HIR

As shown in Fig 3, when the concentration of As stress in the PDA was 40 mg/L, the weight of the hyphae cultivated was slightly lower than that of the control group ($p > 0.01$), and the hyphae growth was partially inhibited with an HIR of 2.5%. The same results also appeared in another As-stress group at 80 mg/L ($p > 0.01$) with an HIR of 4.6%. However, the phenomenon that the hyphae color became dark was observed in the two treated groups. The results illustrated that the presence of 40 or 80 mg/L As did not significantly inhibit the hyphae growth on the agar plates ($p > 0.01$), with part of the hyphae aging.

### Morphological analysis

Under no stress (Fig 4a and 4d), the surface of the hyphae was rough with wrinkles and relatively thicker than those in the As-treated groups. In addition, as shown in Fig 4a, there was a

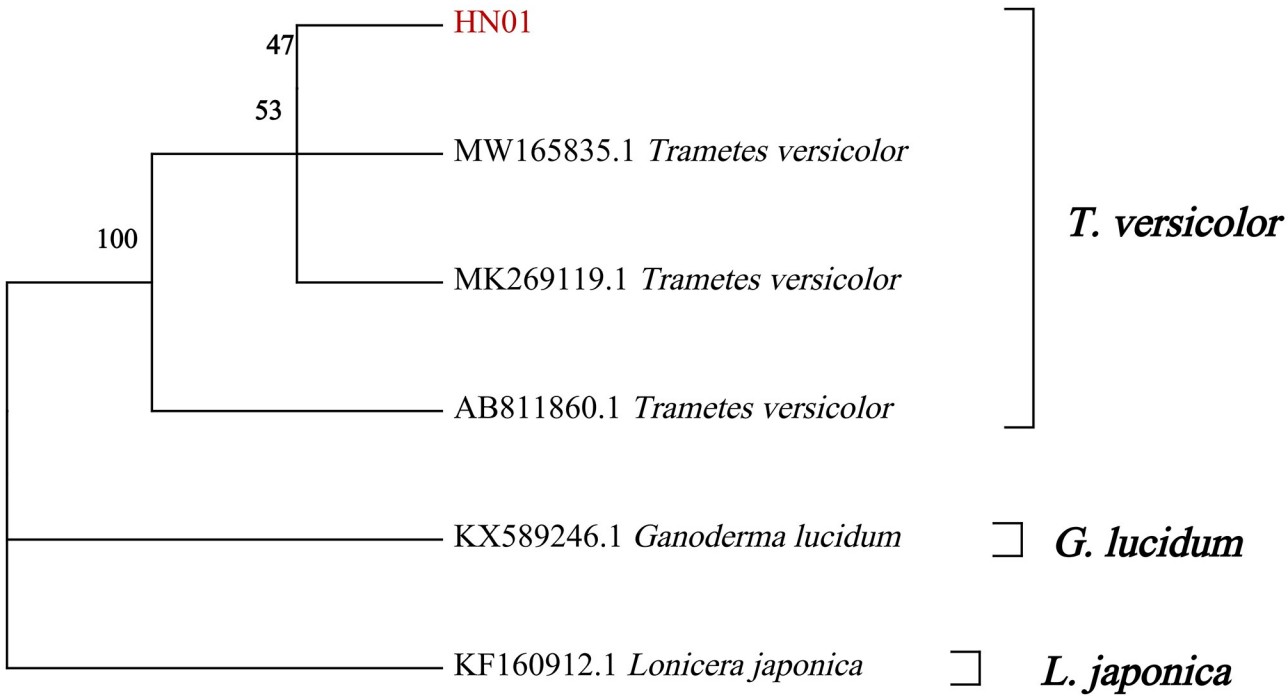

**Fig 2. Phylogenetic tree of HN01 strain based on ITS sequences.**

network structure with dense arrangements near the normal hyphae with many branches, indicating the nutrient transport system of the hyphae was well developed. Under As stress at 40 mg/L, the hyphae network became loose and thin (Fig 4b) with fewer wrinkles (Fig 4e). When the concentration of As stress in the PDA reached 80 mg/L, the hyphae was characterized by having tuberculate, curved branches that were decreasing, as shown in Fig 4c and 4f.

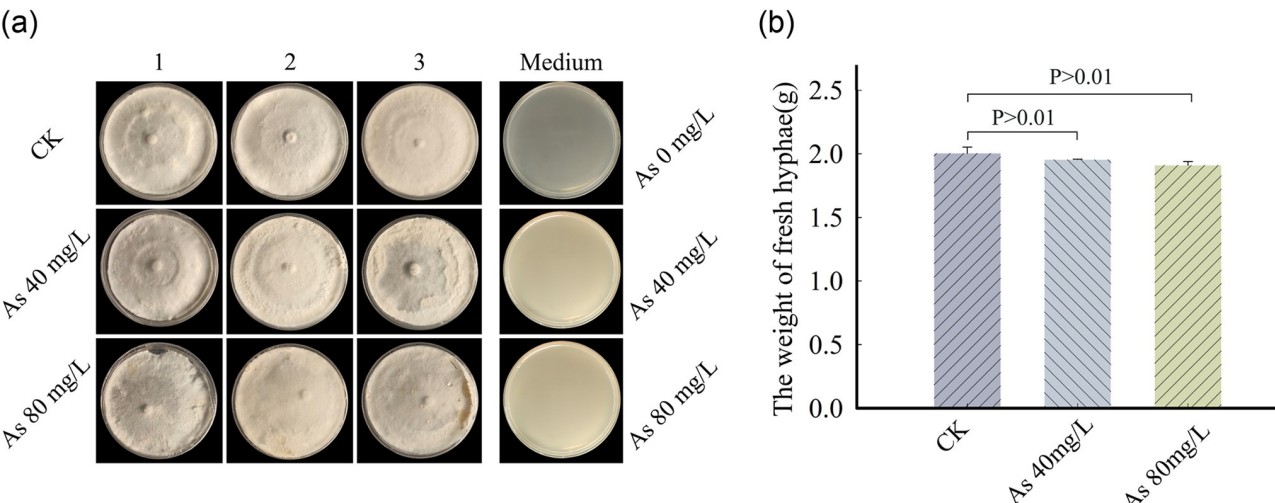

**Fig 3. Effect of As stress on the hyphae growth of *T. versicolor* (n = 3).** CK represented the control group. 3 PDA plates without growth (Medium-As 0 mg/L, Medium-As 40 mg/L and Medium-As 80 mg/L) were placed for observing the darkness clearly.

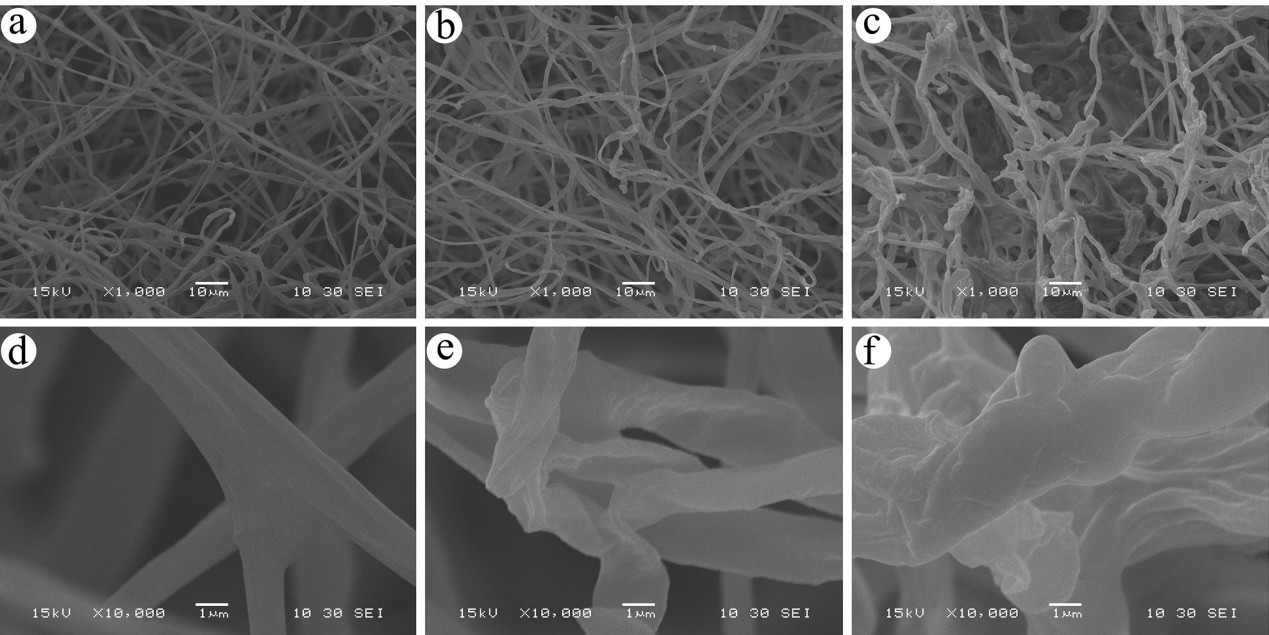

**Fig 4. Morphology analysis of *T.versicolor* hyphae by SEM.** a: hyphae in the control group (×1000); b: hyphae under As stress at 40 mg / L (×1000); c: hyphae under As stress at 80 mg / L (×1000); d: hyphae in the control group (×10000); e: hyphae under As stress at 40 mg / L (×10000); f: hyphae under As stress at 80 mg / L (×10000).

## Effects of As stress on the activities of antioxidants in the *T. Versicolor* hyphae

The results from Fig 5A and 5B indicated significant differences in the activities of CAT and SOD between the control group and the As-stress group at 80 mg/L ($p < 0.01$), whereas no statistical significance ($p > 0.05$) was observed between the control group and the As-stress group at 40 mg/L, thereby showing that the 80 mg/L As could trigger a rise in the activities of CAT and SOD. It is clear that the levels of GSH, which is a key factor against As toxicity [34], significantly increased ($p < 0.001$) in the As-treated groups in contrast to the control group (Fig 5C). However, the activity of MDA did not change significantly ($p > 0.05$, Fig 5D). In our study, the activities of CAT, SOD, and GSH in the As-stress group at 80 mg/L were 1.10, 1.09, and 20.47 times that for the control group respectively, indicating that the oxidative damage caused by As was relieved by SOD, CAT, and GSH.

## As contents in the *T. versicolor* hyphae and the BCF to As

According to Table 1, the natural As concentration in the dry hyphae from the control group was approximately 0.01 ± 0.02 mg/kg, and this strain showed no As accumulation capacity with a BCF < 1. However, when incubated for 10 d under 40 mg/L As stress, the hyphae survived, and the As content increased sharply to 275.47 ± 5.9 mg/kg. In addition, this strain showed a paradoxical As accumulation capacity at a BCF of 13.57. When incubated under an 80 mg/L As concentration, the As content in the hyphae reached 502.10 ± 7.30 mg/kg with no significant growth inhibition and a BCF of 11.25. Combined with the result from section 3.2, this wild strain may be one of the few mushrooms with simultaneous As accumulation and tolerance capacities.

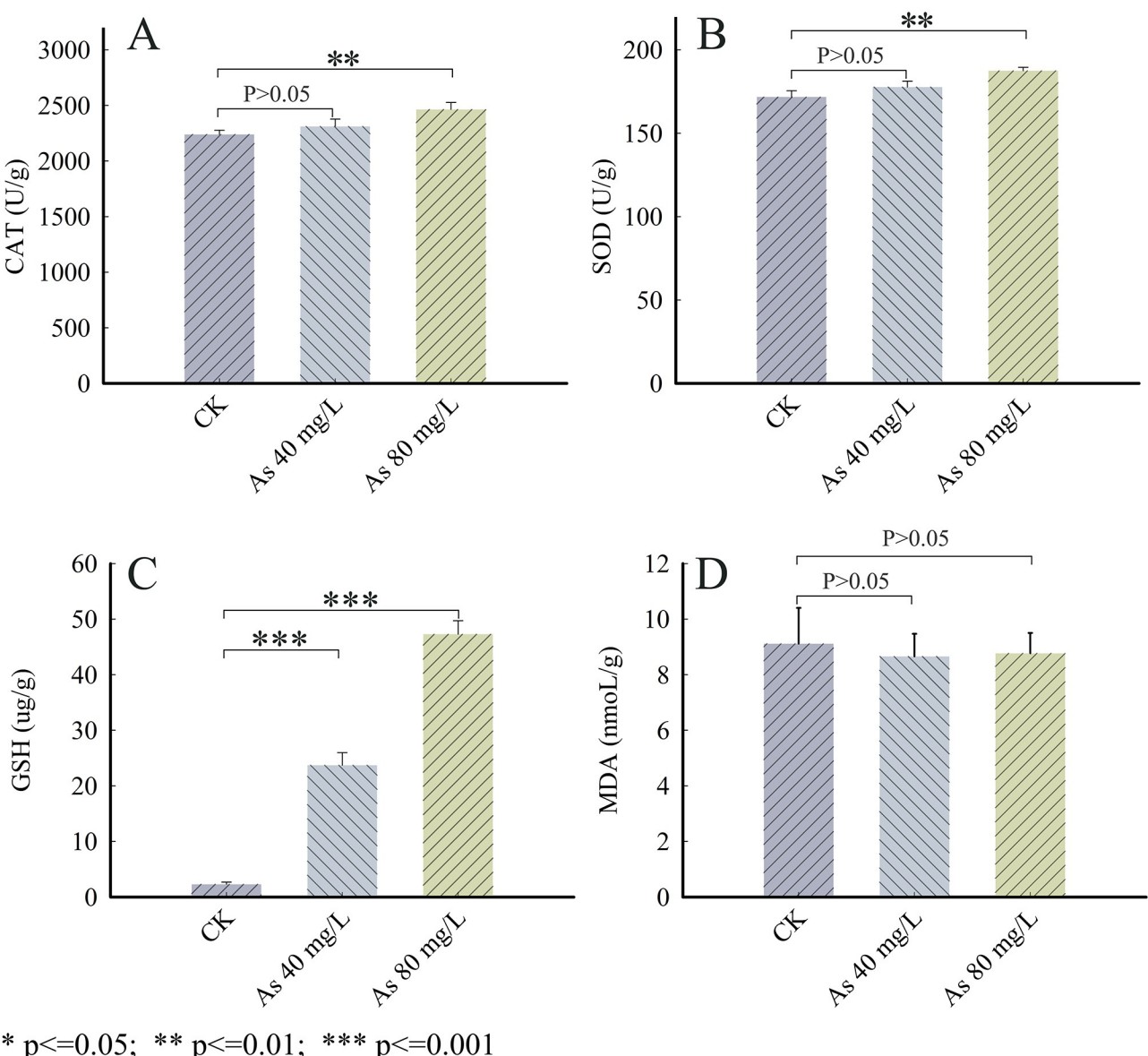

**Fig 5. Effects of arsenic stress on antioxidant activities in *T. versicolor* hyphae.** A: CAT activity (U / g); B: SOD activity (U / g); C: GSH content (ug / g); D: MDA content (nmol / g); * represented statistical difference ($p < 0.05$); * * represented significant difference ($p < 0.01$); * * * represented extremely significant difference ($p < 0.001$); CK represented the control group.

### As speciation in the hyphae of *T. versicolor* under As stress

The analytical performance of the HPLC CP-MS method was validated by determining the linearity, limits of detection, average recovery, and relative standard deviation (RSD), as shown in Table 2. A five-point calibration was performed in the concentration range of 0–300 μg/L. The linear equations were calculated, and good correlation coefficients were found in all cases for $As^V$, $As^{III}$, MMA, DMA, AsC, and AsB ($> 0.999$). The limits of detection were 1.9, 2.3, 1.8, 2.3, 1.8, and 1.6 μg/kg, respectively. Under the extraction and determination conditions from the methods section, the average recoveries of As species were $> 93.6\%$, and the RSDs

**Table 1. As contents in hyphae of *T. versicolor* and BCF to As (n = 3, mg / kg, dw).**

| Treatments | As-stress content(mg / L) | As content (mg / kg) | | | BCF# |
|---|---|---|---|---|---|
| control | 0 | hyphae | 0.01±0.02 | $C_{h0}$ | <1 |
| | | medium | 0.03±0.01 | $C_{m0}$ | |
| As 40 | 40 | hyphae | 275.47±5.90 | $C_{h40}$ | 13.57 |
| | | medium | 20.30±1.55 | $C_{m40}$ | |
| As 80 | 80 | hyphae | 502.10±7.30 | $C_{h80}$ | 11.25 |
| | | medium | 44.63±2.01 | $C_{m80}$ | |

# BCF to As = $C_h$ / $C_m$; $C_h$ (mg / kg) represented As concentration in the dry hyphae; $C_m$ (mg / kg) represented As concentration in the PDA medium after the hyphae harvested.

**Table 2. Analytical performances, recoveries and extraction efficiency for As species by HPLC ICP-MS.**

| Analytes | Linear range (µg/L) | Linear equation | $R^{2}$* | Limits of detection (µg/kg) | Average recovery (%) | RSD# (%) |
|---|---|---|---|---|---|---|
| As$^{III}$ | 0–300 | Y = 21333*X | 1.0000 | 2.3 | 95.7 | 3.7 |
| As$^{V}$ | 0–300 | Y = 30926*X | 1.0000 | 1.9 | 93.6 | 4.4 |
| MMA | 0–300 | Y = 28882*X | 0.9997 | 1.8 | 102.4 | 5.2 |
| DMA | 0–300 | Y = 29164*X | 0.9999 | 2.3 | 96.3 | 2.6 |
| AsC | 0–300 | Y = 17876*X | 0.9999 | 1.8 | 105.3 | 4.2 |
| AsB | 0–300 | Y = 24646*X | 1.0000 | 1.6 | 98.2 | 2.8 |

* $R^2$ represented the linear correlation coefficient of the linear equation.

#RSD represented the relative standard deviation.

were < 5.2%. This indicated that this method was suitable for the accurate detection of the six As species in the hyphae samples.

Table 3 and Fig 6 show that there were primarily four As species in the hyphae of HN01 in the control group: As $^V$, As $^{III}$, AsB, and MMA. The primary species was As $^V$, accounting for 79.75%, and this was followed by the As $^{III}$ proportion of 8.86%. The ratio of AsB to MMA was the smallest. Under an As stress of 40 mg/L, there were five As species in the hyphae including As $^V$, As $^{III}$, AsB, MMA, and DMA. The order of proportion was the following order (highest to lowest): As $^V$ (88.66%) > As $^{III}$ (1.79%) > AsB (0.71%) > MMA (0.37%)> DMA (0.07%). The result of the As speciation from the As-stress group at 80 mg/L was in agreement with that from the As-stress group at 40 mg/L, with an order of proportion of the following: As $^V$ (95.80%) > As $^{III}$ (1.57%) > AsB (0.78%) > MMA (0.24%)> DMA (0.04%). The HPLC ICP-MS results showed the hyphae from the different treatments all contained a majority of As $^V$, As $^{III}$, AsB, and MMA. Considering As $^V$ was the dominant species in the hyphae with no

**Table 3. As species in the hyphae of *T. versicolor* (n = 3, mg / kg).**

| Samples | Total As | As$^{III}$ | As$^{V}$ | MMA | DMA | AsC | AsB | Extraction efficiency(%) |
|---|---|---|---|---|---|---|---|---|
| Hyphae-control | 0.79±0.10 | 0.07±0.04 | 0.63±0.09 | 0.01±0.01 | nd | nd | 0.01±0.01 | 91.1 |
| Hyphae-As40 | 280.30±5.44 | 5.02±0.34 | 248.52±6.37 | 1.03±0.01 | 0.20±0.04 | nd | 1.99±0.03 | 91.6 |
| Hyphae-As80 | 505.30±7.20 | 7.93±0.15 | 484.10±10.30 | 1.22±0.04 | 0.21±0.04 | nd | 3.92±0.02 | 98.4 |

"nd" represented data could not be determined.

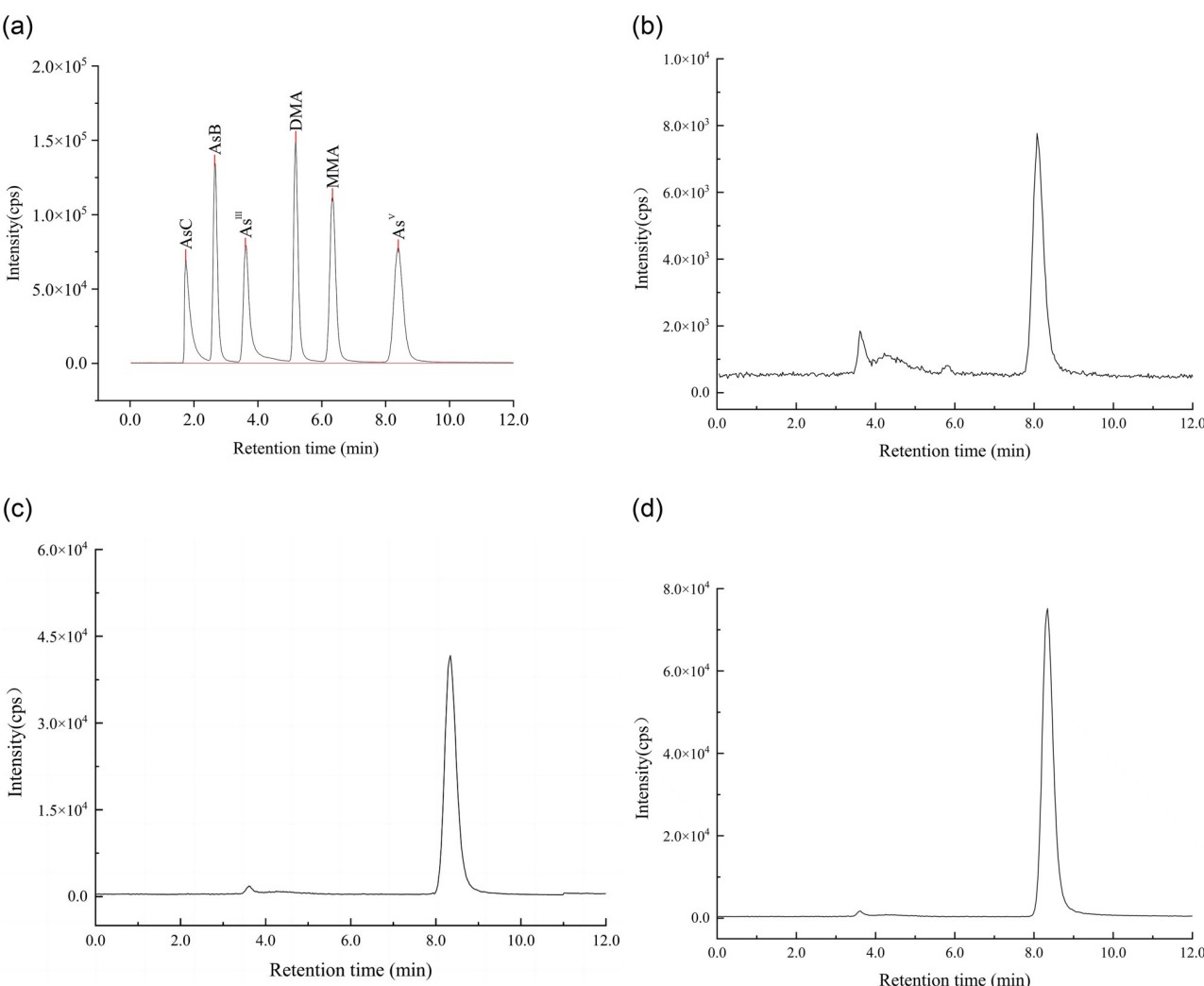

**Fig 6. HPLC ICP-MS chromatograms of As species in standard solution and hyphae samples.** A: chromatograms of six As species with the concentration at 50 μg / L. B: chromatograms of As species in the hyphae of *T. versicolor* from the control group. C: chromatograms of As species in the hyphae of *T. versicolor* from stress group with As III concentration at 40 mg / L. D: chromatograms of As species in the hyphae of *T. versicolor* from stress group with As III concentration at 80 mg / L.

stress or As stress in our research, it seemed that converting As $^{III}$ to As $^{V}$ might be partly connected with detoxification mechanism of *T. versicolor*.

## Correlation analysis

For the correlation heatmap of the four antioxidants, the total As and four As species were analyzed using the Spearman correlation, as shown in Fig 7. Some important correlations were focused on based on the contents of GSH, and the total As and As $^{V}$ increased rapidly with an increase in the As-stress concentration. (1) GSH content: The statistical positive correlation ($p < 0.05$) between the GSH content and the As $^{V}$ content showed that the exogenous As stimulated the GSH and As $^{V}$, and both showed a sharp simultaneous increase in the hyphae. There was also a positive correlation between GSH and the AsB content. (2) Total As content: The positive correlation ($p < 0.05$) between the total As and the As $^{V}$ content could be

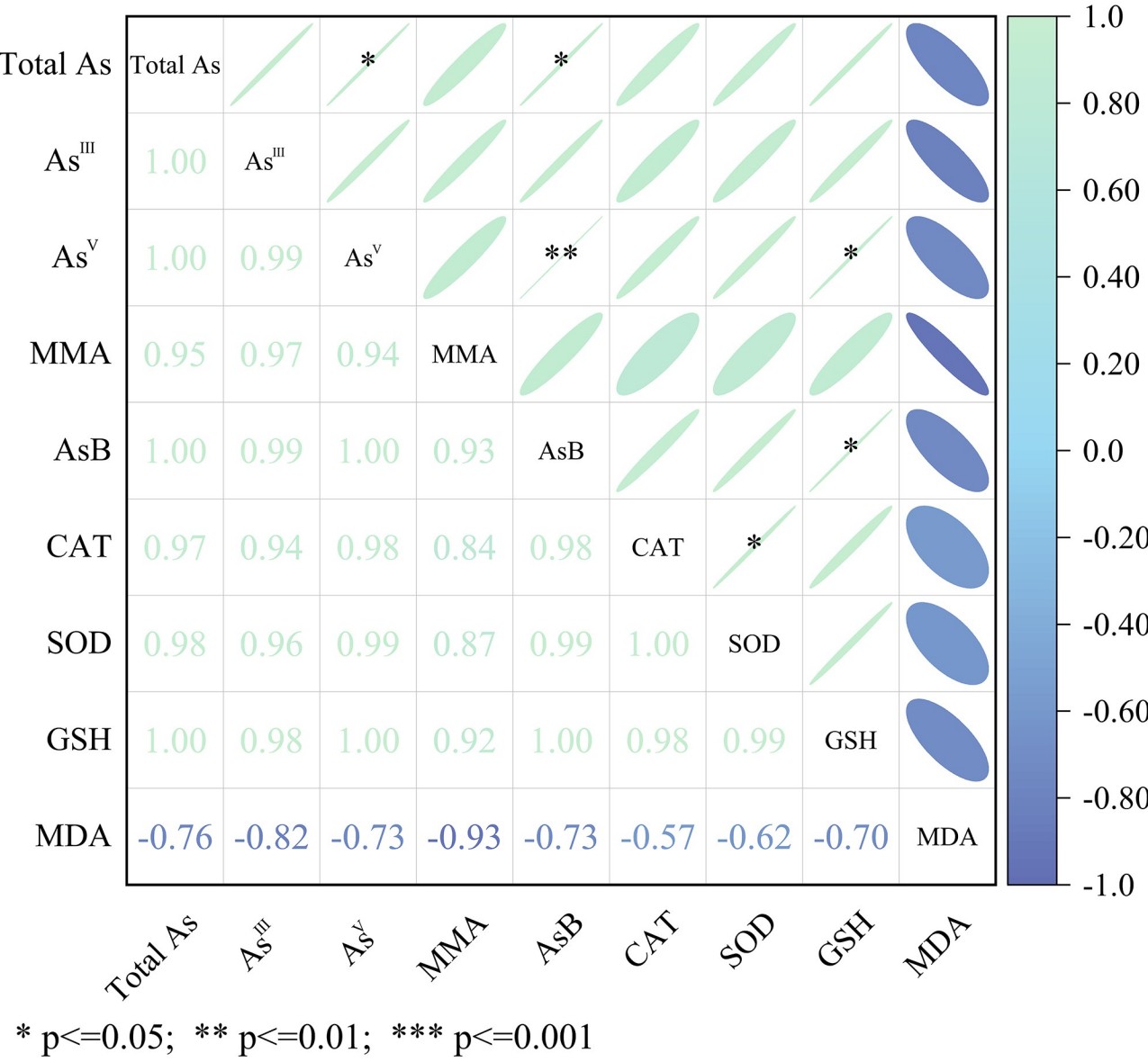

**Fig 7. Correlation heatmap among 4 antioxidants, total As and 4 As species was conducted using Spearman correlation.**

* p<=0.05;  ** p<=0.01;  *** p<=0.001

explained because As $^{V}$ was the dominant species in the hyphae with no stress or As stress. The strain continuously transferred the As $^{III}$ ions from the PDA to the hyphae, arousing an increase in the total As from the hyphae and activating certain detoxification mechanisms to convert As $^{III}$ into As $^{V}$. These mechanisms also might include the conversion of As to AsB because of the positive correlation ($p < 0.05$) between the total As and the AsB content. (3) As $^{V}$. In addition to the positive correlation between As $^{V}$ and the GSH content, there was also a significantly positive correlation ($p < 0.01$) between the As $^{V}$ and the AsB contents. Although the AsB content was correlated with the total As and the As $^{V}$ contents, whether AsB was bio-transformed from As $^{III}$ or As $^{V}$ was unsure in this study. (4) Others: Singh et al. reported increased activity of SOD and CAT under As exposure (5 and 50 μM), suggesting that the

cooperative action of antioxidants was required for a detoxification mechanism under metal or metalloid stress [35]. This could explain the positive correlation between the activities of SOD and CAT ($p < 0.05$).

## Discussion

### Accumulation and tolerance capacities in fungi

Some fungi can accumulate different amounts of As in their fruiting bodies. Organisms that can enrich exceptionally high concentrations of a metal or metalloid element are often thought of as hyper-accumulators. The hyperaccumulation limit for As is 1000 mg/kg [35]. For nearly 30 years, *Sarcosphaera coronaria*, *Laccaria amethystine*, and *Cyanoboletus pulverulentus* have been reported widely as As hyperaccumulating mushrooms [36–38]. In the family Polyporaceae of the phylum Basidiomycota from China, most did not show an enrichment capacity for As, with an As content of 0.0007–2.6 mg/kg dw [25]. The BCF to As indicates the As accumulation capacity in a strain. Generally speaking, the higher the BCF value is, the stronger the enrichment capacity is. A systematic search of mushrooms from 29 different genera was conducted to study the As content and the BCF [39]. The BCF values in most genera were greater than one. The highest BCF was 29 in the fruiting bodies of *Laccaria vinaceoavellanea* with As content of 130–163 mg/kg and an As content of 5.6 mg/kg in soil. This indicated that *L. vinaceoavellanea* was a potential As accumulator. In our study, the BCF was 11.25 in the hyphae of *T. versicolor* with As content of 502.10±7.30 mg/kg and an As content of 44.63±2.01 mg/kg in the PDA medium under an As stress of 80 mg/L. This strain impressed us because of the high As content and strong As accumulation capacity. One prerequisite for a strain to be applied as a bio-accumulator is a strong accumulation capacity, especially under heavy metal polluted conditions. This *T. versicolor* strain would be considered a potential As-accumulator, the same as *L. vinaceoavellanea*. However, As accumulation by macrofungi remains poorly understood.

High tolerance to As is another important prerequisite for fungi to be applied in the field of environmental mycoremediation. The growth of most plants or mushrooms is affected by As toxicity under the condition of a low As concentration, and their growth is inhibited or even die. This concentration range had been reported to be $< 20$ mg/kg in plants [14, 40]. Few organisms can tolerate medium or high concentrations of As with their growth unaffected. Based on this, although some As high- or hyper-accumulating mushrooms have been reported, it was not possible for them to be applied for As removal until their tolerance capacities improved. A *Lycoperdon pyriform* strain that was collected in a polluted environment in Canada was the only macrofungi reported with 1010 mg/kg As and a good tolerance capacity [28]. *Trichoderma sp*., as arbuscular mycorrhizal fungi, also has been extensively exploited in agriculture for environmental mycoremediation because of the good As accumulation and tolerance capacities [21]. HIR represents the tolerance capacity of a metal or metalloid. In our research, although the growth of *T. versicolor* hyphae under As stress was inhibited partially with an HIR of 4.6%, there were no significant differences between the control group and the As-stress group at 80 mg/L ($p > 0.01$). In other words, this wild strain at least tolerated an As concentration at $> 80$ mg / L.

### Antioxidants and detoxification

Plants possess a specific mechanism to prevent reactive oxygen species (ROS) from exceeding toxic threshold levels, and this plays an important role in the acclimation process against metal or metalloid stress. The antioxidant mechanism is comprised of two types of components: enzymatic antioxidants (e.g., SOD and CAT) and non-enzymatic antioxidants (e.g., GSH and MDA). The importance of antioxidants is based on the fact that their increased and/or

decreased levels are generally related to the enhanced or declined stress tolerance of stressed plants [22]. It seems that the phenomenon also applies to As-stressed macrofungi. In our study, with an increase in the As stress concentration, the contents of SOD, CAT, and GSH in the hyphae increased as a response to the oxidative damage caused by As. The activities of the SOD, CAT, and GSH contents depended on the As exposure dose.

GSH is a widely distributed redox active molecule and plays a key role in protecting the membrane from ROS damage. In most cases, heavy metals typically cause a decline in the level of GSH in plant tissues, and it predefines the growing consumption of GSH that is used for the synthesis of phytochelatins (PCs) [41, 42]. However, the GSH content increased with an increase in the As concentration in this study. Approximately 23.71 μg and 47.29 μg of the total GSH were produced per gram of hyphae in response to 40 mg/L and 80 mg/L of As stress, respectively, indicating that GSH biosynthesis was highly inducible by external As stress. Similar observations have also been reported in the ectomycorrhizal fungus *Laccaria bicolor* [34]. According to our results, similar to *L. bicolor*, it can be speculated that GSH biosynthesis is an important mechanism responsible for As detoxification in *T versicolor*.

## As speciation and detoxification

When the HPLC ICP-MS was applied to As species analysis over some years successfully, improving the extraction efficiency becomes challenging. For biological samples, a good method for pretreatment needs to effectively break down the bonds between elements and molecules based on thoroughly disrupting the cell membranes. Strong oxidizing reagents or high temperature conditions can help achieve the complete release and high extraction efficiency (95–100%), but the initial form of As is also changed and the former leads to environmental hazards [33]. The extraction efficiency of As species for mushrooms is 55–104% according to a recent study [25]. However, due to a dense fibrous structure, our preliminary experiments proved that it was not suitable for the dry hyphae of *T. versicolor* to refer to the pretreatment methods above. We explored a freeze-drying method for pretreatment in this study that was applied for treating the hyphae of *T. versicolor*. After the detection of the HPLC ICP-MS, the extraction efficiencies of the As species were all $\geq$ 91.1%. This result indicated that this was a complete process that included freeze-drying pretreatment, microwave-assisted digestion, and the HPLC ICP-MS detection conditions suitable for the As speciation analysis in the hyphae of *T. versicolor*.

It has been previously confirmed that the major As compound in different As-accumulating mushrooms might be different [43]. *Black boletus*, *Suillellus_luridus*, and *Boletus edulis* have been reported to be As-accumulating mushrooms, and the As species were further explored [33, 44]. Komorowicz found that MMA was significantly dominant, and its content was up to 99.4% of the total As in *Suillellus_luridus* and *Boletus edulis* [44]. Chen et al. also found that the primary As species was MMA, which accounted for 93.5% of the total As in *Black boletus* [33]. It appears that MMA was the primary As form in many As-enrichment mushrooms that have been investigated. However, the results of this study showed that the contents of As [V] were 0.63, 248.52, and 484.10 mg/kg in the control group and the two As-stress groups, which accounted for 79.75%, 88.66%, and 95.8% of the total As for *T. versicolor*, respectively. However, the contents of MMA were 0.01, 1.03, and 1.22 mg/kg, respectively. It deserves to be mentioned that we also found 3.92 mg/kg AsB in the As-stress group at 80 mg/L that was converted by inorganic As. Scientists have been inclined to think that AsB is primarily found in seafood [45]. However, Chen et al. found that the content of AsB reached 39.21 mg/kg, and this accounted for 92.7% of the total As in *Lentinus edodes* [33]. However, other scientists have claimed inorganic As is the primary As form in *Lentinula edodes* [46, 47]. These results

demonstrate the different primary As compounds not only in different As-accumulating mushrooms but also in the same mushroom species. It has been speculated that these differences might be connected with the species, stress or no stress, the growing environment, the substrate, or other factors. This conclusion basically agrees with the Simone Braeuer analysis [48]. To our knowledge, this is the first study to demonstrate that As$^V$ is both the dominant compound in the hyphae of *T. versicolor* HN01 regardless of no-stress or As-stress. Researchers believe that most mushrooms have a higher proportion of organic As and a lower inorganic As [49]. These contradictory questions provide entry points for further studies regarding the mechanisms.

Different As species represent distinct levels of toxicity. The most dangerous form is As$^{III}$, which is more toxic than As$^V$. MMA and DMA are considered to be less toxic than inorganic As. AsB is generally considered to be non-toxic [44]. In aquatic systems, some photosynthetic microorganisms accumulate As$^V$ and biotransform it to As$^{III}$ and subsequently to methylarsenic species [18]. It seemed that the situation for *T. versicolor* was opposite to the law of As transformation in aquatic plants. For *T. versicolor*, the As speciation results showed As$^V$ to be the dominant species under no stress or As$^{III}$ stress. This strain accumulated As$^{III}$ and biotransformed it to a majority of As$^V$, AsB, and other methylarsenic species for As detoxification. As shown in Table 3, the contents of As$^V$ in the hyphae from the three groups were evidently higher than that of all the other compounds. We also noticed that a certain amount of AsB accumulated into the hyphae under As stress. This finding motivates further research regarding the source of AsB, since AsB is not made during the vegetative life stage of some reported fungi [43, 50]. This finding combined with the correlation analysis shown in Fig 7 also demonstrates that further research is required to study whether AsB is biotransformed from As$^{III}$ or As$^V$. In this article, the biotransformation of As$^{III}$ to As$^V$, AsB and methylarsenic species (DMA and MMA) are considered to be related to detoxification mechanisms of *T. versicolor*.

There are no studies on As detoxification based on the correlation between antioxidants and As species. We originally integrated data from the antioxidant analyses, the ICP-MS analysis, and the HPLC ICP-MS determination and summarized these into a schematic diagram of the As$^{III}$ detoxification pathways in *T. versicolor*. According to the results shown in Fig 7, the two aspects connected with detoxification were proposed. (1) GSH: GSH is thought to play an important role in As tolerance/detoxification in plants. Inside the cell, As$^V$ is reduced to As$^{III}$ by As-reductase (AR), leading to the conversion of GSH to its oxidized form (GSSG) [18]. This process is necessarily accompanied by the consumption of GSH and a reduction of the As$^V$ content. Many studies have clarified this phenomenon [41, 42]. It is clear that the detoxification pathway above cannot be applied to explain the detoxification in the *T. versicolor* hyphae because the exogenous As$^{III}$ stimulated GSH and As$^V$ to produce a simultaneous sharp increase in *T. versicolor*. Then a hypothesis was proposed that As$^{III}$ is enzymatically oxidized to As$^V$ by As$^{III}$ oxidase secreted by associated microorganisms with the synthesis of GSH after As$^{III}$ is taken up. This hypothesis has received much support from some newly published studies [51, 52]. Kumari and Sanyal claimed that there are a large number of symbiotic microorganisms that secret arsenite oxidase and arsenate reductase in nature. It has been speculated that the content of GSH in the *T. versicolor* hyphae came from the synthesis of GSH but also might have originated from a reduction in the GSH consumption for PCs synthesis [34]. (2) Converting As$^{III}$ to the As$^V$: Due to some facts found in our research, like a large amount of As$^{III}$ with no exogenous As$^V$ in the PDA before incubation and a high concentration of As$^V$ at 484.10 mg/kg in the hyphae after incubation, we can be sure that As$^V$ was converted from As$^{III}$ primarily without exceptions. Some microorganisms have been reported that can secrete As$^{III}$ oxidase in nature, and this can explain this pathway [51, 52]. However, this detoxification

mechanism of converting As $^{III}$ into As $^V$ is not in agreement with previous results obtained from plants and other mushrooms [18, 53]. The detoxification mechanism of *T.versicolor* might involve the biotransformation of As $^{III}$ to As $^V$ accompanied by secretion of GSH.

## Conclusion

This study evaluated As tolerance and accumulation capacities of a wild *T. versicolor* strain under As stress. Some tolerance/detoxification mechanisms have been also generalized in this fungus. Results indicated that *T. versicolor* may be one of the few mushrooms with simultaneous As accumulation and tolerance capacities. The detoxification mechanisms of this strain were involved alleviating the toxicity by increasing the activities of antioxidants, especially GSH, as well as by converting As $^{III}$ into As $^V$ and other less toxic As species. These above which were not in agreement with previous results obtained from plants will require further study and confirmation. *T. versicolor* might be used as a bio-accumulator to deal with As exposure in As-contaminated environments based on its extraordinary As tolerance and accumulation capacities.

## Supporting information

**S1 Raw images.**
(PDF)

## Acknowledgments

We thank LetPub (www.letpub.com) for its linguistic assistance during the preparation of this manuscript.

## Author Contributions

**Conceptualization:** Chenyu Wang.

**Data curation:** Xiaohong Chen.

**Funding acquisition:** Ling Xie.

**Investigation:** Chenyu Wang.

**Methodology:** Xiaohong Chen.

**Resources:** Xiaohong Chen.

**Supervision:** Zhilan Xia.

**Writing – original draft:** Yuhui Zhang, Ling Xie.

**Writing – review & editing:** Ke Xiao, Ling Xie.

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
