## [Decision Letter · Decision Letter 0]

20 Mar 2023

PONE-D-23-04265Antioxidants and speciation changes involved in arsenic detoxification about a wild Trametes versicolor strainPLOS ONE

Dear Dr. xie,

Thank you for submitting your manuscript to PLOS ONE. After careful consideration, we feel that it has merit but does not fully meet PLOS ONE’s publication criteria as it currently stands. Therefore, we invite you to submit a revised version of the manuscript that addresses the points raised during the review process.

We look forward to receiving your revised manuscript.

Kind regards,

Jameel Al-Obaidi

Academic Editor

PLOS ONE

Journal Requirements:

Reviewers' comments:

Reviewer's Responses to Questions

**Comments to the Author**

1. Is the manuscript technically sound, and do the data support the conclusions?

Reviewer #1: Partly

Reviewer #2: Yes

2. Has the statistical analysis been performed appropriately and rigorously? 

Reviewer #1: Yes

Reviewer #2: Yes

3. Have the authors made all data underlying the findings in their manuscript fully available?

Reviewer #1: Yes

Reviewer #2: Yes

4. Is the manuscript presented in an intelligible fashion and written in standard English?

Reviewer #1: Yes

Reviewer #2: Yes

5. Review Comments to the Author

Reviewer #1: Comments to Author(s)

Thank you for submitting the manuscript.

Title: The title is meaningless and needs a complete change.

Abstract:

The abstract lack a proper introduction, purpose, and methodology. These three items must be added to the abstract.

Introduction:

In this section, the authors presented good info about the contamination of As. However, the impact of As contamination on human health and As toxicity in the food chain reaching ultimate consumers (animals and plants) were not mentioned in the “Introduction”.

The lines from 86 to 94 are not necessary and need summarization.

Depending on what is presented in the “Introduction”, the last paragraph should focus on the aims of the study. The aims were not clear.

Materials and Methods:

In general, this section contains many grammar mistakes and sometimes not clear sentences.

Strain, chemicals and medium

The first paragraph should revise, “The T. versicolor HN01 strain was obtained from China Center for Type Culture Collection (CCTCC M 2021010)” without further information. The culture conditions of the strain should mention.

The origin/company of chemicals should mention.

Internal transcribed spacer (ITS) molecular identification and the phylogenetic tree

1) Indicate the forward and reverse primers used for ITS1 and ITS4.

2) Did the authors check the purity of extracted DNA?

Statistical analysis

The p value should be small and Italic.

Results

ITS Molecular identification and phylogeny of the Trametes species

Can the authors demonstrate the purpose of this experiment? The strain used in their research was already isolated, identified, and authenticated by China Center for Type Culture Collection. Is this strain isolated from contaminated area or showed significant tolerance or accumulation of metal contamination?

I think this step is irrelevant with the context or the purpose of the manuscript.

Hyphae growth 213 of T. versicolor and HIR

The figure 3 is not clear, I highly recommend the authors to replace this figure (the plates). The non-differences are quoit obvious, however neither the hypha growth nor the darkness were observed. I prefer to add PDA without growth as blank.

Replace CK group with control group whenever mentioned, including figures and tables.

As speciation in the hyphae of T. versicolor under As stress

The results are very conceived.

Discussion:

Accumulation and tolerance capacities in fungi

This sentence is irrelevant please remove it “As a medicinal mushroom from the family Polyporaceae, whether T. versicolor enriches As remains a secret”.

The included good info about mycoremediation, my suggestion is to summarize it in one paragraph.

Antioxidants and detoxification

“This result further confirmed the general lack of PCs in most fungi”, how can the authors justify this statement? Any possible methods to detect the percentage of PCs in T. versicolor after As treatments?

As speciation and detoxification

Please remove this sentence “The HPLC ICP-MS detection method was utilized to learn from herbal samples and rice [42, 43]”.

Line 407: The sentence “In this article, the biotransformation of Asv to AsIII, AsB and methylarsenic species” is incorrect. It should be “the biotransformation of AsIII to Asv.

Some hypotheses about As detoxification pathways in T. versicolor

There are some misleading suggestions about the pathway of As III detoxification to As V, which is the main finding in this manuscript. The authors proposed 3 different pathways.

Main Issue: There are many mistakes in describing the conversion event, As III is always biotransformed to As V. The authors should focus on this main pathway which involves the biotransformation of As III to As V accompanied by overexpression of GSH.

The other pathways were not supported by the results in this manuscript, so I highly recommend deleting them and invest bioinformatic tools to propose a clear pathway for As III detoxification. This recommendation also includes Fig. 8.

Where is the conclusion?

Reviewer #2: 1. The paper is easy to follow by the reader. However, I may require some comments on the following issues.

i. Title: The title is good.

ii. Abstract: This section was well-written and easy to understand.

iii. Introduction: Adequate.

iv. Material and Methods: excellent

v. Results: Good

vi. Discussion: The discussion of the study is well written and cover the important topics.

vii. Conclusion: Excellent.

6. PLOS authors have the option to publish the peer review history of their article (what does this mean?). If published, this will include your full peer review and any attached files.

Reviewer #1: **Yes: **Ali Z. Al-Saffar

Reviewer #2: **Yes: **Dr Ahmed A. M. Elnour

---

## [Author Response · Author response to Decision Letter 0]

24 Mar 2023

Attched in a submitted file naming " response to reviewers".

---

## [Decision Letter · Decision Letter 1]

25 Apr 2023

PONE-D-23-04265R1Arsenic(III)-induced oxidative defense and speciation changes in a wild Trametes versicolor strainPLOS ONE

Dear Dr. ling xie,

Thank you for submitting your manuscript to PLOS ONE. After careful consideration, we feel that it has merit but does not fully meet PLOS ONE’s publication criteria as it currently stands. Therefore, we invite you to submit a revised version of the manuscript that addresses the points raised during the review process.

We look forward to receiving your revised manuscript.

Kind regards,

Jameel Al-Obaidi

Academic Editor

PLOS ONE

Journal Requirements:

Reviewers' comments:

Reviewer's Responses to Questions

**Comments to the Author**

1. If the authors have adequately addressed your comments raised in a previous round of review and you feel that this manuscript is now acceptable for publication, you may indicate that here to bypass the “Comments to the Author” section, enter your conflict of interest statement in the “Confidential to Editor” section, and submit your "Accept" recommendation.

Reviewer #1: All comments have been addressed

Reviewer #2: All comments have been addressed

2. Is the manuscript technically sound, and do the data support the conclusions?

Reviewer #1: Yes

Reviewer #2: Yes

3. Has the statistical analysis been performed appropriately and rigorously? 

Reviewer #1: Yes

Reviewer #2: Yes

4. Have the authors made all data underlying the findings in their manuscript fully available?

Reviewer #1: Yes

Reviewer #2: Yes

5. Is the manuscript presented in an intelligible fashion and written in standard English?

Reviewer #1: Yes

Reviewer #2: Yes

6. Review Comments to the Author

Reviewer #1: Comments to Author(s)

Thank you for submitting the manuscript and adherence to the notes.

The authors made great efforts to improve the manuscript, kindly consider the following notes.

Title: No comments.

Abstract:

No comments

Introduction:

Perfectly improved.

Materials and Methods:

Strain, chemicals and medium

First line: obtained by issue ----- I think they meant “by tissue”.

“Pre-dried to achieve constant weight” something missing with this sentence, needs rephrase.

Results

Hyphae growth 213 of T. versicolor and HIR

I'm still not convinced with Fig. 3 and the author’s response; the authors did not replace this Fig. with a better one. Replacing the Fig. is a vital issue, since it’s the ground base of the manuscript.

Discussion:

No Comments.

Reviewer #2: Dear Editor,

I have reviewed the revised submitted article, which investigates the arsenic tolerance and detoxification mechanisms of the wild fungus Trametes versicolor HN01 under arsenic stress. The study is well-structured and presents valuable insights into the potential use of T. versicolor as a bio-accumulator for dealing with arsenic exposure in contaminated environments.

The methodology employed in the study is sound, and the results provide clear evidence of the fungus's ability to tolerate high levels of arsenic and effectively detoxify the metalloid. The findings on the role of antioxidants, especially glutathione, in detoxification, as well as the conversion of As III into As V and other less toxic species, are both novel and significant.

Although the results differ from previous plant studies, the authors recognize this discrepancy and suggest further research to confirm their findings. This honesty and transparency in the discussion only serve to strengthen the article's credibility.

Based on the quality of the research, the relevance of the findings, and the potential applications of T. versicolor in environmental remediation, I highly recommend accepting this article for publication. The study will undoubtedly contribute to the existing body of knowledge on the subject and spark further research in the field.

Sincerely,

Dr. Ahmed A. M. Elnour

7. PLOS authors have the option to publish the peer review history of their article (what does this mean?). If published, this will include your full peer review and any attached files.

Reviewer #1: No

Reviewer #2: **Yes: **Dr. Ahmed A. M. Elnour

---

## [Author Response · Author response to Decision Letter 1]

27 Apr 2023

A file named Response to Reviewers was attached.

---

## [Decision Letter · Decision Letter 2]

9 May 2023

Arsenic(III)-induced oxidative defense and speciation changes in a wild Trametes versicolor strain

PONE-D-23-04265R2

Dear Dr. Ling Xie,

We’re pleased to inform you that your manuscript has been judged scientifically suitable for publication and will be formally accepted for publication once it meets all outstanding technical requirements.

Kind regards,

Jameel Al-Obaidi

Academic Editor

PLOS ONE

Additional Editor Comments (optional):

Reviewers' comments:

Reviewer's Responses to Questions

**Comments to the Author**

Reviewer #1: All comments have been addressed

Reviewer #2: All comments have been addressed

2. Is the manuscript technically sound, and do the data support the conclusions?

Reviewer #1: Yes

Reviewer #2: Yes

3. Has the statistical analysis been performed appropriately and rigorously? 

Reviewer #1: Yes

Reviewer #2: Yes

4. Have the authors made all data underlying the findings in their manuscript fully available?

Reviewer #1: Yes

Reviewer #2: Yes

5. Is the manuscript presented in an intelligible fashion and written in standard English?

Reviewer #1: Yes

Reviewer #2: Yes

6. Review Comments to the Author

Reviewer #1: (No Response)

Reviewer #2: The authors have diligently considered all feedback from the reviewers to enhance the quality and rigor of their research. As suggested, they made essential modifications, updates, and clarifications to the manuscript, ensuring all concerns and recommendations are thoroughly addressed. This process not only improves the overall presentation of the study but also bolsters the credibility of the results and conclusions. By integrating the valuable input from the reviewers, the authors have honed their work, leading to a more robust, well-reasoned, and refined manuscript that makes a substantial contribution to the knowledge in their specific field.

7. PLOS authors have the option to publish the peer review history of their article (what does this mean?). If published, this will include your full peer review and any attached files.

Reviewer #1: No

Reviewer #2: **Yes: **D. Ahmed Elnour

---

## [Editor Report · Acceptance letter]

18 May 2023

PONE-D-23-04265R2 

Arsenic(III)-induced oxidative defense and speciation changes in a wild *Trametes versicolor* strain 

Dear Dr. Xie:

I'm pleased to inform you that your manuscript has been deemed suitable for publication in PLOS ONE. Congratulations! Your manuscript is now with our production department. 

Kind regards, 

on behalf of

Dr. Jameel Al-Obaidi 

Academic Editor

PLOS ONE